# Non-Aneurysmal Perimesencephalic Subarachnoid Hemorrhage: A Literature Review

**DOI:** 10.3390/diagnostics13061195

**Published:** 2023-03-22

**Authors:** Iulian Roman-Filip, Valentin Morosanu, Zoltan Bajko, Corina Roman-Filip, Rodica Ioana Balasa

**Affiliations:** 1Department of Neurology, “George Emil Palade” University of Medicine, Pharmacy, Sciences and Technology, 540136 Targu Mures, Romania; 2Department of Neurology, “Lucian Blaga” University of Sibiu Faculty of Medicine, 550169 Sibiu, Romania

**Keywords:** nonaneurysmal, perimesencephalic, venous structures, cisterns, pathophysiology, diagnosis, tomography, fisher scale, Hunt and Hess scale, outcomes, clinical features, treatments

## Abstract

Spontaneous non-aneurysmal subarachnoid haemorrhage (NAPMSAH) (addressing point 1) is a relatively rare occurrence in clinical settings as it is rarely misdiagnosed and usually involves a significantly better prognosis than the classical aneurysmal pattern. We hereby focused on a comprehensive analysis of this distinct pathological entity with the purpose of analysing possible pathophysiological entities, outcomes and treatment options involving this diagnosis with a focus on demographical, epidemiological and clinical data. The clinical setting includes focal neurological signs related to the anatomical structures, while computer tomography followed by tomographic angiography are the most common diagnosis tools, with a typical hyperdense lesion involving the midbrain, fourth ventricle and subthalamic areas without an angiographic correspondent, such as an aneurysmal pathology. Further investigations can also be used to highlight this diagnosis, such as interventional angiography or magnetic resonance imaging. Given the rarity of this condition and its relatively better prognosis, treatment options usually remain conservative. In the present review, the main characteristics of NAPMSAH are discussed.

## 1. Introduction

Subarachnoid haemorrhage is a significant cause of morbidity and mortality worldwide. The most common non-traumatic causes of spontaneous subarachnoid haemorrhages are usually associated with aneurysmal ruptures, while other non-aneurysmal causes are a considerably rarer encounter.

The perimesencephalic subarachnoid haemorrhage (PMSAH) is a relatively rare type of subarachnoid bleeding and can be shortly described as a build-up of blood in the proximity of the interpeduncular cistern, a particular anatomical region that contains several structures of vital importance including the posterior cerebral artery, the superior cerebellar artery and the third cranial nerve [1].

Therefore, given the complexity of the interpeduncular region and the relative proximity to the suprasellar and pontocerebellar cisterns, the specific location of bleeding appears to be variable; a typical position in the vicinity of the cerebral peduncles or caudally, such as the prepontine region, with the extension of blood into the suprasellar cistern or into the proximal Sylvian fissures [2,3].

One of the particularities of this type of deep brain haemorrhage lays in its incidence, about 0.3–0.5 cases per 100,000 persons [2]. PMSAH also accounts for about 5–10% of subarachnoid haemorrhages and more than 30% of all non-aneurysmal subarachnoid haemorrhages [3]. Compared to the aneurysmal pattern of subarachnoid bleeding, the isolated variant is associated with an encouraging, often self-limited, outcome [4].

Despite the fact that digital subtraction angiography represents the most accurate device that can be used to exclude aneurysmal causes, the usage of CT angiography has increased as it is a specific investigation tool that can be safely used to emphasize or exclude aneurysmal patterns [4].

Of all perimesencephalic subarachnoid bleeding cases, more than 15% have an unknown cause as it cannot be distinguished with our current diagnostic tools [5]. Moreover, as up to 12 percent of all posterior circulation aneurysmal locations are within the interpeduncular cistern, angiographic imaging studies, such as direct subtraction angiography (DSA) or computed tomography angiography (CT angiography) appear (addressing point 2) to be the only high-specificity investigation that can point to a distinctive cause [6].

After exclusion of aneurysmal pathology, the cause of the bleeding usually remains unknown as there are a lot of possible incriminating factors as well as other possible diagnoses that should not be overlooked including dural venous sinus thrombosis, cerebral vascular malformation or trauma [7,8].

The location of most of the arachnoid vessels that profoundly supply the midbrain is within the Virchow–Robin spaces; the enlargement of this space can increase the risk of bleeding within the area, as is seen in several cases, such as repeated head trauma or possible high-volume arteriovenous malformations [9,10,11].

The radiographic pattern of PMSAH is relatively distinct, with the haemorrhage centred anterior to the midbrain or pons, with or without extension of blood around the brainstem, into the suprasellar cistern, or into the proximal Sylvian fissures [12,13].

## 2. Epidemiology

The incidence of non-traumatic subarachnoid haemorrhage appears to be relatively low, with an estimated annual incidence of about 6 cases per 100,000 cases worldwide.

The majority of non-traumatic subarachnoid haemorrhage cases are caused by sacular aneurysmal rupture and are characterized by a poor outcome with a mortality rate that can exceed 50% [14].

While the typical risk factors for stroke are related to aneurysmal rupture, it appears that non-traumatic non-aneurysmal bleeding is not generally linked to the typical risk patterns for subarachnoid bleeding [15].

Anke Zhang et al. conducted a retrospective study involving 273 cases of aneurysmal subarachnoid haemorrhage (aNSAH) (addressing point 3) between 2013 and 2018 with the purpose of comparing the long-term outcome for angiography negative perimesencephalic and non-perimesencephalic bleeding patterns. The study aimed to evaluate the outcome of those patients within 3 and 12 months following the diagnosis. The following results were shown: only 8.7% of the patients diagnosed with perimesencephalic bleeding experienced a worsening of clinical status. In the case of non-perimesencephalic bleeding patterns, the study concluded that clinical worsening was reported in 14.3% of those cases. The result has, therefore, underlined the fact that the perimesencephalic bleeding pattern is generally associated with a better prognosis [16].

Kim et al. conducted another retrospective analysis involving a total of 51 cases of non-aneurysmal perimesencephalic subarachnoid haemorrhage, focusing on the medium- and long-term outcome following the diagnosis. The cases were analysed using modified Rankin Scale (mRS) and the study pointed out that there is a general pattern of positive outcomes, with approximately 80.3% of the patients having significant improvement within 6 months following the diagnosis. Moreover, the study identified several factors that were generally associated with a better prognosis, including a good mRS at onset, absence of early hydrocephalus, non-Fisher type 3 or higher bleeding model, and a relatively short-term haemorrhage (with blood resorption within 7 days from onset). The overall conclusion of the study is that the absence of angiographic findings is typically associated with a better medium-term prognosis [17].

## 3. Risk and Causality Factors

A certain number of records aimed to identify possible risk factors that can be associated with non-aneurysmal perimesencephalic subarachnoid haemorrhage (NAPMSAH), such as hypertension, smoking, diabetes, or alcohol abuse. There are multiple studies that conclude that up to 31% of patients with NAPMSAH are smokers [17,18], while about 2/3 of the patients diagnosed with aneurysmal SAH are known smokers [19]. Gupta and his collaborators interestingly underlined an association between diabetes and NAPMSAH in a series of six records, concluding the fact that NAPMSAH is more often associated with diabetes (17%) compared to aneurysmal subarachnoid haemorrhage (aSAH), with an overall prevalence of diabetes in those cases accounting for only 10–12% [20].

There appears to be no specific causality relationship between alcohol consumption and NAPMSAH; on one hand, three studies associated NAPMSAH with alcohol consumption in as low as 10% to as much as 30% of the cases [21,22], while on the other hand, a study developed by Caeiro and Santos describing the neuropsychiatric impact of subarachnoid haemorrhages suggests that there is no link between alcohol abuse and symptomatic haemorrhage as no patient included in the research had a history of alcohol abuse [23]. Ildan and his collaborators have studied a certain number of prognoses and prognostic factors in a series of 29 cases of NAPMSAH; their findings concluded that there was no case of alcohol abuse highlighted within their research [24].

## 4. Pathogenic Sequence and Causes

Given the anatomical configuration of the vessels surrounding the cerebral peduncles and cerebellum, we can differentiate several causal factors.

Increased pressure in the intracerebral venous circulation has also been linked to NAPMSAH [25]. There is a number of cases that describe certain anomalies in the venous circulation, including the straight sinus, jugular vein or the vein of Galen [26,27], linking them to NAPMSAH. Increased intracranial pressure usually leads to the enlargement of the vessels and possibly their rupture, causing bleeding. The outcome in this case is generally benign as the cardinal symptom is usually a non-specific headache and the treatment is usually symptomatic [28].

Disturbances in deep brain blood flow appear to be a valid causative factor for NAPMSAH as there is a complex venous system surrounding the midbrain, including the internal cerebral veins, basal vein of Rosenthal and the dural sinuses. The embryological classification of the variants of midbrain drainage consists of three types of possible venous architecture: normal continuous (type A), normal discontinuous (type B) and primitive variant (type C). During embryological development, the basal vein of Rosenthal is formed by the longitudinal fusion of three primitive veins as they initially drain in the tentorial sinuses, but as the development progresses, those vessels start to obliterate as their drainage shifts towards the Galenic system [29,30]. Various failures of anastomosis between first and second primitive veins results in type B and type C drainage, respectively, while type A is considered ideal [24]. The study performed by Irene C. and collaborators aimed to underline the involvement of defective venous drainage in PMSAH using non-contrast computer tomography and concluded that a type C malfunctioning drainage is present in at least one hemisphere in 29 out of 52 patients (53%) while normal drainage was seen in only 7.3% of the cases. The deep venous drainage for the posterior circulation was analysed [31]. Leakage of blood following the rupture or fissure of an arteriovenous fistula is a classical cause of subarachnoid bleeding. There is a particular subtype of arteriovenous malformation linked to NAPMSAH, mainly concerning the arteries of Davidoff and Schechter. Those arteries are a branch of the posterior cerebral artery and supply the parts of the falx cerebri and of the tentorium [32].

The arteries of Davidoff and Schechter are not commonly seen on conventional angiographic studies unless they suffer an abnormal enlargement, such as in the case of dural arteriovenous fistulas [33,34], meningiomas or infratentorial tumours. Anatomically, they can originate from the superior surface of the P1-2 junction or the proximal P2 segment and they usually have a posterolateral trajectory between the fourth cranial nerve and the posterior cerebellar artery within close proximity to the free edge of the tentorium [35]. Together with the anterior meningeal branches, they are responsible for the blood supply of the tentorium. They typically unite with the contralateral arteries at the edge of the falx cerebri piercing the dura and ascend cranially towards the superior sagittal sinus. The abnormal connection with superior sagittal sinus is regarded as the cause of this arteriovenous malformation [32,34,35]. The amount of blood leaking can either be minimal, therefore requiring no particular treatment [33], or substantial, therefore requiring endovascular treatment [34].

Perforating basilary artery aneurysms can also be a cause of SAH and, more often than not, conventional angio-CT scans or subtraction angiographies fail to assess any aneurysmal dilatation. These types of aneurysms usually produce mild-to-moderate bleeding, often similar in pattern with NAPMSAH and following a clinical course similar with other SAHs. In order to highlight such an entity, repeated angiograms are required, with only a dozen cases reported in current literature (less than 60). A recent case reported by Mithun et al. has successfully pointed out a perforating aneurysm in a 62-year-old patient with clinical SAH after four consecutive angiographs; the patient was subsequently effectively treated using overlapping stent placement [36].

## 5. Clinical Factors, Long-Term Functional and Cognitive Outcomes

Patients diagnosed with NAPMSAH generally present with milder symptoms than a typical aneurysmal SAH. Loss of consciousness is unusual. The most commonly reported symptoms include a suddenly installed headache, meningeal irritation (in the meaning of neck stiffness), Kernig and Brudzinski signs, photophobia, nausea and vomiting.

The study conducted by Şahin, S. et al. underlined the fact that the vast majority of patients diagnosed with NAPMSAH, respectively 93%, were classified as Hunt–Hess Grade I or II, whilst less than 7% of the patients were of Grade III or higher, further highlighting the lower severity of this type of bleeding [35].

Furthermore, another study carried out by Schwarz et al. reports no patient as having a Hunt–Hess Grade III or higher upon presentation. However, up to 20.8% of the participants were reported to develop some form of loss of consciousness. Focal neurological deficits included diplopia, seventh nerve paralysis and medulla involvement, dysphagia, involvement of respiratory nucleus and alteration of breathing patterns [37].

In order to assess the outcome of the patients, the most commonly used scales include the modified Rankin scale and the Glasgow Outcome Scale [38].

Various studies conclude the fact that up to 97% of patients with this diagnosis are expected to live normally following discharge. All the patients included in a study performed by Matsuyama and collaborators were reported to have an outstanding short-term evolution, being able to return to their previous activity within short notice [39].

Given the relatively small size and the fact that there is no acute decline in the cerebral blood supply, alterations of the mental status of these patients is relatively rare.

There is generally some form of deficit related to the local anatomical structures being directly involved in the bleeding process, rather than diffuse involvement of the brain parenchyma [25].

There is a general consensus that most patients are able to return to daily activities, including work, following discharge. However, up to 30% of the patients involved in a study were reported to have a severe daily activity impairment following their discharge, including the loss of work capacity [38,40].

Other non-specific long-term complaints include persistent headache, attention and short-term memory impairment and even anosmia [40,41]. Surprisingly, the life expectancy of the patients appears not to be influenced [42,43].

The cognitive function of the patients is also of important interest as concerns arise as to whether the mental capacity of the patients is affected or not. Several studies dedicated to evaluating the mental outcome of those patients have been elaborated. Firstly, Godefroy and his collaborators used the Modified Mental Status Scale to demonstrate the fact that there is little-to-no cognitive impairment on long-term follow-up for these patients [44].

Given the relatively close proximity of the midbrain to crucial white matter structures, including the thalamo–limbic connecting pathways, possible damage to those white matter structures could lead to cognitive impairment and memory loss. In that regard, a study was performed, using the means of diffusion tensor imaging, showing that diffuse damage to these structures was demonstrated; furthermore, this damage was linked with some cognitive impairment in the subjects [45,46]. However, the precise mechanism of white matter injury remains unknown. Some evidence points to a multitude of haemorrhage-induced small-sized emboli [47].

## 6. Diagnosis and Treatment Approach

The conventional diagnostic approach focuses on imagistic findings via CT\as well as CT angiography to expose possible aneurysms. The plain CT has a high blood specificity, it is time and cost effective and is mostly preferable to other imagistic studies. Moreover, the specificity of this investigation can reach 100% in certain optimal conditions [48]. In order to precisely diagnose a subarachnoid bleed, another comprehensive diagnostic tool is CSF analysis via lumbar puncture, which has to be correctly timed and performed [49].

CT angiography correlating with plain-CT provides useful aetiological and prognostic information as the negativity of the first correlated with the negativity of the latter imply a milder prognosis [50]. In addition, after thorough confirmation using the more subacute oriented MR angiography, this can possibly imply a non-aneurysmal haemorrhage. Additionally, we have to underline the fact that several false-negative middle cerebral artery aneurysms can appear on MR angiography [51].

Digital subtraction angiography is another vital method to detect underlying vascular problems and highlighting possible aneurysmal dilatations of the vessel wall. Radio-opaque structures, such as bones, are eliminated via digital methods from the image (hence the term subtraction), therefore enabling the user to accurately view the blood vessels. In this regard, a meta-analysis was performed by Potter and his collaborators involving 252 cases of perimesencephalic subarachnoid haemorrhage who benefited from at least two such procedures (including the ones performed at follow-up visits). The results of this study highlighted the importance of this procedure as the cases that were included in the study fulfilled both the clinical criteria for NAPMSAH and the imagistic criteria as well, with no aneurysmal dilatation being shown on serial procedures (at diagnosis and 10 days follow-up). The importance of DSA in excluding aneurysmal causes of bleeding is considered vital in the diagnosis of NAPMSAH as the capacity to exclude surrounding tissues and reduction of imaging artifacts provides a clearer imagistic study compared to classical CTA. Most authors strongly recommend that this procedure should be performed within the acute period in order to safely exclude an aneurysmal cause of perimesencephalic bleeding [52], to clearly assess the importance of DSA in NAPMSAH.

There has been increasing usage of intracranial vessel wall MR imaging (VWI) in a considerable amount of intracranial vascular pathologies. The main advantage of vessel wall imaging lies in the ability to directly visualise the vessel wall and to be able to possibly provide a cause for idiopathic NAPMSAH. The study performed by J.M Couthinho et al. aimed to use this method as a diagnosis tool in NAPMSAH. The study included 11 patients diagnosed with the pathology and highlighted a series of vessel-wall changes, including wall enhancement and abnormalities of the basilary artery vessel wall, but with little impact on patient management [53,54].

It is important to note the fact that the diagnosis of the perimesencephalic non-aneurysmal pattern of bleeding is classically a diagnosis of exclusion as one has to firstly safely exclude aneurysmal bleeding and the blood has to be specifically located in the perimesencephalic and prepontine cisterns. Notably, blood is rarely to be seen within the ventricles [51].

Though less often emerging than in the case of aneurysmal bleeding, there have been several notable reports of hydrocephalus following non-aneurysmal bleeding [55].

Despite the fact that up to half of the patients admitted for an aneurysmal type bleeding develop a certain electrolyte balance disturbance, the most common imbalance seen is by far hyponatremia. The most common cause of hyponatremia following subarachnoid bleeding is related to an antidiuretic hypersecretory syndrome caused by diffuse hypothalamic injuries following relatively large magnitude bleedings. There is little evidence in the literature linking the NAPMSAH to hyponatremia, as the size of bleeding is relatively small, therefore, rarely causes diffuse subcortical damage [56].

The meta-analysis conducted by Jin P. et al. focused on describing the incidence of seizures for angiogram-negative subarachnoid haemorrhage by grouping a 612 patients and revealing the fact that the perimesencephalic haemorrhage had a much lower incidence of seizures compared to non-perimesencephalic haemorrhage, with only four reported cases of seizures compared with 16 cases with non-perimesencephalic haemorrhage, also demonstrating a lower risk association in this case [57].

Regarding the treatment perspectives, treatment for NAPMSAH is usually conservative, including dozens of symptomatic approaches, though there have been several cases where clipping a causative arteriovenous fistula was shown to improve the prognosis; those cases have been exceptional. Several invasive treatments have also been performed in these cases, but they were mostly related to complications, such as hydrocephalus, by placing a ventricular shunt; however, a permanent shunt was placed in no more than 13–13.5% of all cases [57,58].

## 7. A Detailed Overview of Complications

### 7.1. Vasoactive Complications

Even though this clinical entity has a relatively low incidence of complications and usually a favourable outcome, there is a certain percentage of cases that develop vasospastic reactions, usually ranging from 3–20% [59,60]. The general outcome following this complication is usually variable and highly dependent on several factors, the most important of those being the quantitative amount of haemorrhagic volume. The precise quantification of this volume can be achieved using a conventional computed tomography slice and multiplying it by the thickness of the haemorrhagic area involved. A correct appreciation of this volume can also be considered with the prediction of other long-term complications, including hydrocephalus and seizures [61].

The precise degree of vasospastic contraction is highlighted via direct subtraction angiography (DSA) [62]. Furthermore, the severity of the vasospasm can be appreciated according to the narrowing of the vessel: >66% in case of severe vasospasm, between 33 and 66% for moderate and lower than 33% in case of a mild vasospasm [63]. Ultrasonographic methods can also be implemented as to the degree of vasospastic reaction. Therefore, Transcranial Doppler has proven itself to be relatively useful, especially in assessing larger vessel involvement. In this regard, the mean blood velocity is constantly measured as it can be considered significant in case the mean value exceeds 120 cm/s or an increase from the basal value of more than 50 cm/s [64]. Various studies have pointed out that the most logical vessel that is usually involved is the proximal portion of the basilar artery, whilst one study that followed the ultrasonographic evolution of those patients pointed out the fact that the onset and the peak of the vasospastic reactions occurred during the first week following the onset of the haemorrhage, with a significant decrease in the intensity in the following week and eventual cessation within 3 weeks [65].

When vasospastic reactions occur, they are generally asymptomatic, whilst delayed neurological deficits occur in less than 1% of the cases [66]. The risk of vasospasm has been traditionally appreciated using the Fisher Computer–Tomographic scale developed in 1980 with a grading of 1–4, with grade 1 being the lowest and grade 4 the highest probability of vasospasm: 1—absence of blood, grade 2—diffuse subarachnoid haemorrhage that does not exceed 1 mm and no clots, grade 3—localized clots and layers of blood that exceed 1 mm without ventricular involvement, grade 4—subarachnoid haemorrhage with intraventricular bleeding [67]. Later, the Barrow Neurological Institute Scale was established with the aim of more precisely predicting vasospastic outcomes with five grades as follows: 1—no blood; 2—SAH ≤ 5 mm thick; 3—SAH 5–10 mm thick; 4—SAH 10–15 mm thick; and 5, SAH > 15 mm thick [68]. Various studies assessing the Barrow Scale were performed on NAPMSAH patients with a mean scale of 1–3 points, therefore, classifying the vasospastic risk in this situation as low [69,70].

Preventive measures for this particular complication were previously achieved via the “Triple H” therapeutical principle involving haemodilution, hypertension and hypervolemia, while the current emphasis and therapeutical options are centred upon hypertension and cerebral angioplasty with intraarterial injection of hypertensive medication [71].

### 7.2. Hydrocephalus and Fluid Build-Up

Build-up of blood within close proximity of the main cerebrospinal fluid (CSF) drainage sites can lead to an obstruction of those drainage pathways, causing an abnormal accumulation of CSF; therefore, enlarging the ventricular system leading to increased intracranial pressure and focal neurological deficits.

The pathophysiological sequence of hydrocephalus following subarachnoid haemorrhage is complex and involves multiple mechanisms. Blood leakage in the subarachnoid and leptomeningeal space leads to inflammation and subsequent fibrous build-up in this space, therefore, exercising a compressive effect on the venous drainage sites preventing efficient evacuation of the CSF, resulting in a “communicant” form of hydrocephalus with no objective sign of ventricular obstruction [72].

Another interesting pathological theory appears to be linked with vasospasm of the anterior choroidal artery by producing stenosis at the site of the third ventricle and the aqueduct of Sylvius, therefore, producing a “non-communicating” form of hydrocephalus. The vasospastic effect combined with the effective compression exercised by the effective blood volume contribute to the acute phase of hydrocephalus [73].

There are several factors that can alter the risk of acute hydrocephalus following this NAPMSAH, including the bleeding pattern and the estimative value of blood involved. A study developed by Kang and his collaborators demonstrated the necessity of ventricular drainage in a significant number of patients diagnosed with NAPMSAH, with the total number of those patients getting close to 25% in this particular study. Moreover, clinical severity appears to be statistically correlated in those cases with a higher incidence of shunt placement and a strong correlation between diffuse bleeding and increased necessity of intraventricular shunt placement [74].

Regarding the management of hydrocephalus, the vast majority of cases appear to follow a benign course with a significant portion having spontaneous resolution or lumbar punctures [75]. A higher rate of shunt placement was also shown to be correlated with a higher Fisher grade of bleeding [69,75].

## 8. Our Clinical Experience and General Outcomes

Throughout a period of 10 years, there were seven cases of NAPMSAH that were diagnosed in our clinical establishment using conventional diagnostic techniques, including CT and various angiographic methods, such as conventional angiography and DSA. Out of the seven diagnosed patients, one died during hospitalization 2 days after the diagnosis, while the other six patients survived and were eventually discharged with a modified Rankin Scale scores (mRS) (at the moment of discharge) varying from 1 to 4. Unfortunately, there was no data regarding the long-term outcome of those patients as there was no follow-up visit. The mean Rankin score (mRS) of our patients following their discharge was assessed at three points.

Treatment received by these patients included Nimodipine, cerebral depletions and analgesia, while clinical features included neck stiffness, extraocular muscle palsies and internuclear ophthalmoplegia. The patients are listed in Table 1.

## 9. Discussion and Conclusions

Angiography-negative perimesencephalic haemorrhage is a relatively rare type of subarachnoid haemorrhage with a more promising clinical outcome and with a slightly reduced long-term disability rate. NAPMSAH is associated with milder symptoms compared to aneurysmal SAH and the clinical outcome appears to be also significantly better in that regard. Although the risk factors linked to this pathological entity are comparable to those associated with aneurysmal bleeding, the aetiology remains relatively unknown as there are several possible causative factors mostly linking its occurrence to several embryological venous drainage anomalies. The diagnosis of this pathology is rather difficult as it requires angiographic expertise to exclude small-size aneurysms; the most used tools include conventional DSA, CT angiography and plain CT scans [76].

Concerning the complications, these are relatively rare and follow a benign course requiring only symptomatic treatment; invasive procedures are more often the exception than the rule. The occurrence of complications appears to be closely linked to the size of the haemorrhage, while the risk of complications can be assessed by referring to imagistic scales such as the Fisher scale. The long-term follow-up and long-term prognosis appears to be generally favourable in this case as long-term complications appear to be related to a degree of decline in the cognitive function, with little to no residual focal neurological deficits [76].

In conclusion, NAPMSAH is a distinctive pathology that appears to be caused by malfunctions in the drainage of the deep brain venous system. It can be safely diagnosed via conventional CT angiography or DSA and treatment options are usually conservative with an emphasis on complications. The main difference from the classical aneurysmal haemorrhage lays in the more benign evolution and general outcome.

## Figures and Tables

**Figure 1 diagnostics-13-01195-f001:**
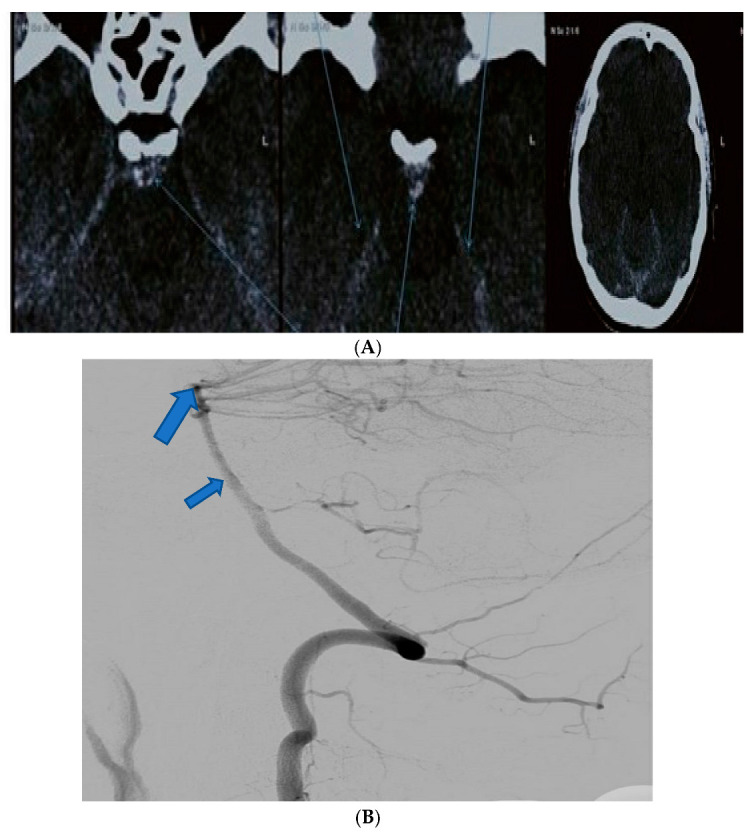
(**A**) The case of a 41-year-old male without any significant clinical history who presented in our clinical establishment complaining of diplopia. Axial native CT highlighting blood near the cerebellar tentorium and prepontine cistern (arrows). (**B**) The same case. Digital subtraction angiography showing no aneurysmal dilatation of the basilary artery (small arrow), cerebellar or proximal posterior cerebral artery (large arrow).

**Figure 2 diagnostics-13-01195-f002:**
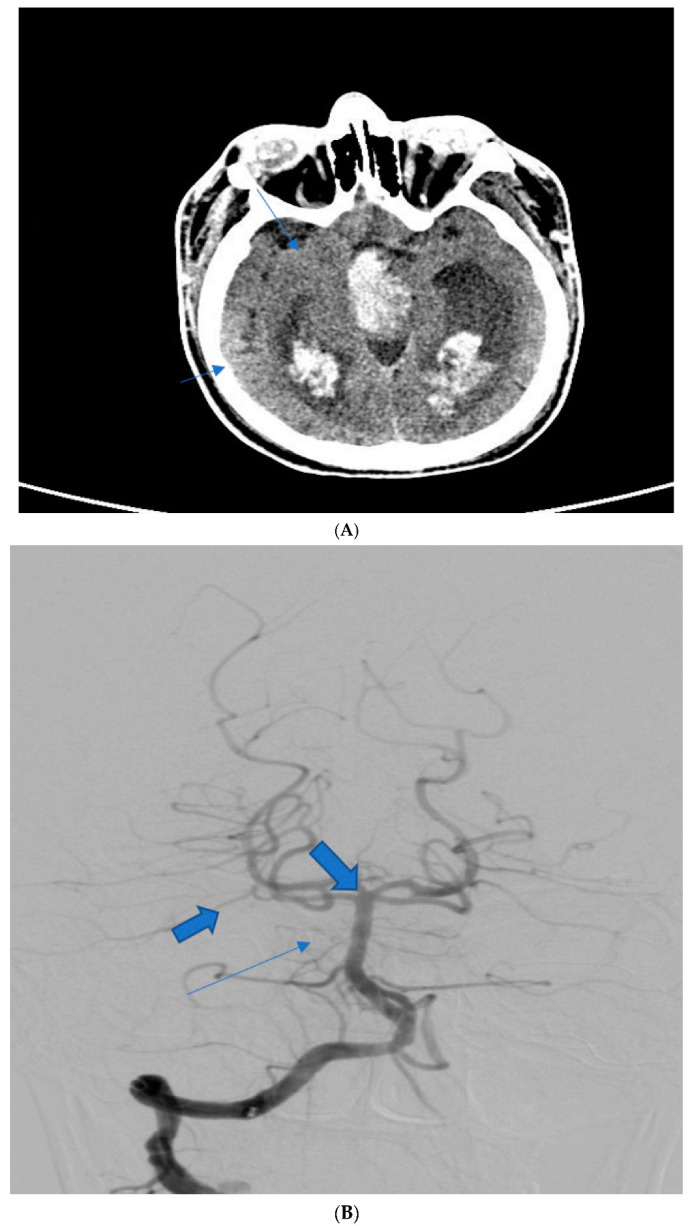
(**A**) The case of a 61-year-old male who was brought to the emergency department with altered mental status and tetraplegia. Native CT scan reveals a massive perimesencephalic haemorrhage with extension to the lateral ventricles with a mFisher scale of 4 points. The patient received depletive therapy as well as nimodipine with an unfavourable outcome as he died one day following admission. (**B**) The same case of the 61-year-old male after digital subtraction angiography was performed showing no aneurysmal dilatation of the basilar (small arrow), posterior cerebral and superior cerebellar arteries (larger arrows).

**Table 1 diagnostics-13-01195-t001:** An overview of the patients diagnosed and treated in our institution.

Patient Age	Patient Sex	Presenting Symptoms	Diagnosis Method	Hunt Hess Scale at Diagnosis	Modified Fischer Scale at Diagnosis	Complications	Rankin Scale at Discharge	Given Treatment
54	F	Diplopia, headache	Native CT scan, CTA	I	I	Mild vasospasm, mild hydrocephaly	2	Nimodipine
61 ^1^	M	Flaccid tetraplegia, altered Mental status	Native CT scan, DSA	IV	IV	Severe vasospasm, hydrocephalus	6 (deceased)	Nimodipine, mannitol
37	M	Headache, diplopia	Native CT scan, DSA angiography	II	II	None	3	Nimodipine
48	F	3rd nerve palsy, pupillary anomalies	Native CT scan, CTA	II	II	Mild vasospasm	2	Nimodipine
41 ^2^	M	Headache, diplopia	Native CT scan, DSA	III	III	Mild vasospasm	3	Nimodipine, mannitol
59	M	3rd nerve paresis, pupillary anomalies, hemiplegia	Native CT scan, DSA angiography	V	IV	Hydrocephalus, moderate vasospasm	4	Nimodipine, mannitol
44	F	Right internuclear ophthalmoplegia	Native CT scan, CTA	III	I	None	1	Nimodipine

Abbreviations: CT scan—computer tomographic scan, CTA—CT angiography; DSA—digital subtraction angiography. ^1,2^ See Figure 1 and Figure 2.

## Data Availability

Not applicable.

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
