# Peer review of "Non-Aneurysmal Perimesencephalic Subarachnoid Hemorrhage: A Literature Review"

_diagnostics, 2023, doi:10.3390/diagnostics13061195_

Round 1

Reviewer 1 Report

This is a literature review article on non-aneurysmal perimesencephalic subarachnoid hemorrhage. This article contains eight sections, from introduction to conclusions, and cites 80 references. The authors also present their patients and images.

There are several points to be concerned.

General comments: The manuscript needs to be reorganized. Too many paragraphs are observed in a section. Many paragraphs contain only one sentence. The content of this manuscript is a bit messy, more like summary notes for a journal review. The terminology used in the manuscripts varies and confuses the readers. English editing is required.

Title: Please correct the spelling “subarachnoid”.

Introduction:

1.     Line 43: “about 0.3-0.5 cases per 100.000 patients”. Is it “per 100,000 patients” or “per 100,000 persons age 18”?

2.     Line 43: Please spell out PMSAH when first mentioned in line 32 and use PMSAH in the rest of the text.

Epidemiology

3.     Line 69: Please use “100,000” instead of “100.000”. Please also explain 6 cases per 100,000 cases - which kind of cases?

4.     Line 68-75: Please combine these three sentences into one paragraph and edit the English.

5.     Line 76-82: It’s a long sentence.

6.     The terms are confusing and some are not identical throughout the manuscript. Please spell out “ANSAH” in line 76, mRS in line 86, NAPSAH in line 96, aSAH in line 101, NCCR in line 134, NAPMSAH in line 139, NPMASH in line 160, CSF in line 206, LP in line 206, NPMH in line 231, and NAPMH in 233.

7.     Some of the statements in [Epidemiology] were discussing risk factors.

Pathogenic sequence and causes

8.     Line 142-150: It’s difficult to understand these sentences. What structures did the authors mean about “they”, dural arteriovenous fistulas or the arteries of Davidoff and Schechter?

Clinical factors, long term functional and cognitive outcomes

9.     Line 157: Please specify symptoms of meningeal irritation.

10.   Line 166: Please specify the neurological deficits resulting from medulla involvement.

Diagnosis and treatment approach

11.  The presence of SAH can be confirmed by CT. The etiological screening of SAH can be achieved by CT angiography or MR angiography. The gold standard for evaluation of possible aneurysm or abnormal vascular lesions is digital subtraction angiography. The comparison of the diagnostic accuracy of CTA and DSA for non-aneurysmal SAH need to be addressed, particular in patients with CTA-negative SAH.

A detail overview of complications

12.  Line 257-259: The diagnostic criteria of vasospasm by transcranial Doppler need to be more clearly defined. In the cited reference (Ref. 67 Nesvick et al. 2019), vasospasm was considered when the maximal TCD velocities was >120 cm/s. While according to the reference cited in the reference 67 (Ref. 12 Rabinstein et al. 2004), mean arterial velocities >120 cm/s on the anterior, middle, or posterior cerebral arteries were indicative of vasospasm.

13.  Line 269: “grade 1”

14.  Line 273: How many levels, 5 or 6, are established on the BNI SAH Grading Scale?

Our clinical experience and general outcomes

15.  Usually, figures from the same patient are combined into one figure, separated by Figure 1A, 1B, 1C... Brain CT findings can be grouped in Figure 1A. the findings of DSA, including anteroposterior and lateral views, can be grouped in Figure 1B. Could the authors possible provide images of the patients with fatal outcome in Figure 2?

16.  The quality of Figures and arrows need to be improved.

17.  Line 312-313: “…using conventional diagnostic techniques including CT and CT angiography”. The diagnostic technique shown in Figure 1 is DSA, not CT angiography.

18.  The authors have performed a comprehensive review of NAPMSAH and attempted to share their patients. They need to report the rate of NAPMSAH in their series. The authors are supposed to report their patients in tabular from following their recommendations, including the presenting symptoms and the Hunt-Hess Grade, the Fisher CT Scale or Barrow Neurological Institute Scale of SAH, the diagnostic tools to screen the aneurysm, the complications, the treatment, and the discharge outcomes.

19.  Seven patients were diagnosed as NAPMH. Why the outcomes were reported only in 5 patients? The discharge mRS ranged from 1 to 4. Given that patients with NAPMH may have better outcomes, a detail mRS needs to be addressed and compared with patients with aneurysmal PMSAH.

20.  Line 320: There seems to be a few words missing at the end of the sentence.

Author Response

I have managed to provide a revised version of the article adressing most of the points of concern that were highlighted. I will also be submitting the article to an English review shortly. Regarding point 11, I didn't quite understand the complaint, and regarding point 15, I have managed to only add a native CT scan. The other points have been generally adressed as i have provided an extensive correction and noted the points that were supposed to be adressed. If there is any underlying problem, I look forward to hearing from you and further indications.

Reviewer 2 Report

I’ve appreciated your efforts of reviewing the literature on this topic that can have a clinica impact

I have some comments 

-please detail the acronyms (ie NAPMSAH) in the text the first time you use them

-in the “epidemiology “ section you present some papers and works from other authors that in my opinion should be better included in the “discussion “ section

-in the “diagnosis “ section you should update the bibliography and include “new” diagnostic tools such as for example MR vessel wall imaging and another pathological entity like basilar perforator aneurysms, that are an overlooked even if rare cause of PMSH

-in the clinical experience you say that you encountered 7 cases whereas further on you talk about 5 diagnosed cases. What about the other 2? It would also be interesting to know the prevalence of sah (total amount of sah in the same time interval)

-please anonymize the picture and assess the position of the arrows

-in the “reference “ section: ref 7 lacks some details. Ref 10 and 12 are the same paper, you should not repeat it

Author Response

Thank you very much for your reply. I have managed to adress the subsequent complaints as the article has been modified accordingly. I have also additionally provided supplementary patient details in the section regarding our clinical experience and adressed the bibliographic sources requiring additional attention.

Further discussions regarding the perforant basilary artery aneurysms and MR vessel imaging have also been updated in the articles at points 5 and 3 respectively. I will also add the article to a comprehensive english review in order to improve the quality of the language.

I will also be looking forward to adressing any further problems.

Round 2

Reviewer 1 Report

The authors have made some modifications in this version. However, there are still many typos and incorrect acronyms in the text that confuse the readers and must be improved. The role of CT angiography and DSA in the diagnosis of NAPMSAH need to be further clarified.

1.      Abstract line 1: Spontaneous nonaneurysmal “perimesencephalic” subarachnoid hemorrhage “(NAPMSAH)”; line 17: “computed tomography”.

2.      Lines 51-52: Please use the same terms of angiographic studies throughout the manuscript, such as digital subtraction angiography (DSA) and computed tomography angiography (CT angiography). The authors can give abbreviations here and use these abbreviations in the rest of the text (for instance, “conventional angio-CT scans or subtraction angiographies” in Lines 188, 310).

3.      Line 81: Please spell out ANSAH and delete its abbreviation. If it is “aneurysmal SAH”, please add “(aSAH)” and use aSAH in line 120.

4.      Is “et all” a correct term? The most commonly used term would be “et al.”

5.      Line 91: Please add “(NAPMSAH)” just after its full term and use NAPMSAH in line 117.

6.      Line 115: Please delete “(added new paragraph concerning the risk factors)”.

7.      Lines 118 and 124: Are “addressing point 5” and “point 5aSAH” unnecessary words in the sentence?

8.      Line 120: Please add “(aSAH)” after aneurysmal SAH and delete “aneurysmal subarachnoid hemorrhage” in line 124. Replace “aneurysmal SAH” with “aSAH” in lines 198 and 422.

9.      Lines 140 and 142: “PNSAH”

10.  Line 158: “NCCT”

11.  Lines 250-251: Please replace “computer tomography” with “CT”.

12.  Line 253: Please correct “In order td o precisely…”.

13.  Lines 254-255: No abbreviation is necessary for lumbar puncture.

14.  Please explain why CT angiography but not DSA is recommended for screening vascular problems in patients with NAPMSAH.

15.  Line 286-288: Please replace “PMH” with “PMSAH” and replace “NPMH” with “non-PMSAH”.

16.  Line 290/355/360/362/372: “NAPMH”

17.  Line 317: Please describe “a peak systolic velocity” or “a mean velocity” of 120 cm/s.

18.  Line 342: Please delete “cerebrospinal fluid”.

19.  Line 378: The mean “mRS” of our patients….

20.  Table 1: “modified Rankin Scale at discharge”, “Mannitol”. Does AngioCT mean CT angiography? If yes, please replace it with CTA and explain the abbreviations of CTA and DSA in the Table note.

21.  Figure 1: The brain CT should be carried out earlier than the angiography. The figure legend of original Figure 1A and 1B are the same. May I suggest merge the fist and second images of Figure 1B and the second image of Figure 1C into Figure 1A, and change the original Figure 1A (DSA) into Figure 1B? The legend of Figure 1A can be shortened as “ …and prepontine cistern (arrows)” and the legend of Figure B can be shortened as “…….digital subtraction angiography showing no aneurysmal dilatation of the basilar artery (small arrow), cerebellar or proximal cerebral artery (large arrow)”. Please correct “computed tomography” in Figure 1 legend. Please also note that vascular study of this 41-year-old male patient was not DSA in Table 1.

22.  Figure 2: Please provide the images of CT angiography in Figure 2B.

23.  Line 421: “NAPMSH”

24.  Line 439: Please replace “direct subtraction angiography” with “DSA”.

25.  Abbreviations: Please reconfirm the necessity of “PMH” and “LP” and add “aSAH” and “DSA”. CT-computed tomography; DSA-digital subtraction angiography

26.  Please provide a clean version of the PDF file during submission.

Author Response

Thank you for your comments.

I have managed to provide a revised version of the article according to the points suggested in this revision. The clean PDF version and clean Word version will be subsequently added.